# Transcriptional Regulation of *HMOX1* Gene in Hezuo Tibetan Pigs: Roles of WT1, Sp1, and C/EBPα

**DOI:** 10.3390/genes11040352

**Published:** 2020-03-26

**Authors:** Wei Wang, Qiaoli Yang, Kaihui Xie, Pengfei Wang, Ruirui Luo, Zunqiang Yan, Xiaoli Gao, Bo Zhang, Xiaoyu Huang, Shuangbao Gun

**Affiliations:** 1College of Animal Science and Technology, Gansu Agricultural University, Lanzhou 730070, China; wangw@st.gsau.edu.cn (W.W.); yangql0112@163.com (Q.Y.); xkh34567@163.com (K.X.); wangpf815@163.com (P.W.); luoruirui628@163.com (R.L.); yanzunqiang@163.com (Z.Y.); gxl18892@163.com (X.G.); zhangb1662@163.com (B.Z.); huanghxy100@163.com (X.H.); 2Gansu Research Center for Swine Production Engineering and Technology, Lanzhou 730070, China

**Keywords:** *HMOX1* gene, promoter, transcriptional regulation, Hezuo Tibetan pig

## Abstract

Heme oxygenase 1 (*HMOX1*) is a stress-inducing enzyme with multiple cardiovascular protective functions, especially in hypoxia stress. However, transcriptional regulation of swine *HMOX1* gene remains unclear. In the present study, we first detected tissue expression profiles of *HMOX1* gene in adult Hezuo Tibetan pig and analyzed the gene structure. We found that the expression level of *HMOX1* gene was highest in the spleen of the Hezuo Tibetan pig, followed by liver, lung, and kidney. A series of 5’ deletion promoter plasmids in pGL3-basic vector were used to identify the core promoter region and confirmed that the minimum core promoter region of swine *HMOX1* gene was located at −387 bp to −158 bp region. Then we used bioinformatics analysis to predict transcription factors in this region. Combined with site-directed mutagenesis and RNA interference assays, it was demonstrated that the three transcription factors WT1, Sp1 and C/EBPα were important transcription regulators of *HMOX1* gene. In summary, our study may lay the groundwork for further functional study of *HMOX1* gene.

## 1. Introduction

Tibetan pigs are typical high-altitude pig breeds living on the Qinghai–Tibet Plateau, and are vital to the lives of Tibetan people [1]. The main producing areas of Hezuo Tibetan pigs are located in the southwestern of Gansu province in Gannan Tibetan Autonomous Prefecture, on the northeast edge of the Qinghai–Tibet Plateau, with an average elevation of about 3000 meters [2]. Hezuo Tibetan pigs belong to a group of Tibetan pigs, which can adapt to the harsh plateau environment and feeding conditions, mainly relying on grazing for their livelihood [3]. Genomic analyses found that there were different patterns of selection between domestic Duroc pigs and Titetan wild boars. Tibetan pigs have a higher proportion of lineage-specific genes, these specific genes were related to vascular smooth muscle contraction, disease resistance, and chemokine signaling pathway, which may reflect that Tibetan wild boar experienced natural selection to adapt to harsh environments [4]. Genome-wide SNP markers confirmed that the divergent evolution between Chinese and Western pigs, the interpopulation linkage disequilibrium, is much longer in Western pigs compared with Chinese pigs. In the genomic research to identify the candidate loci between Tibetan pigs and lowland pigs, several genes in Tibetan pigs are likely important for genetic adaptation to high altitude [5]. Previous research has shown that the adaptability of Tibetan pigs at high altitudes is significantly different from that of other domestic pigs. Tibetan pigs have evolved physiological characteristics to adapt to hypoxia in the plateau, such as a thicker alveolar septum, more developed capillaries [4], and larger and stronger hearts [5]. Therefore, an in-depth study of Tibetan pigs will help us understand the hypoxic adaptability of plateau species [6]. 

Heme oxygenase (HO) is a stress-inducing enzyme that catalyzes heme to produce free iron, carbon monoxide (CO), and biliverdin [7]. Heme oxygenase 1 (*HMOX1*), one of the main members of the HO family, has a variety of cardiovascular protective functions [8,9,10] and plays an important role in anti-inflammatory, anti-apoptotic, and antioxidant activities among others [11,12,13]. *HMOX1* exerts protective effects under stress conditions mainly through the active gas CO and antioxidant bilirubin produced by its decomposition [14,15]. In addition, as a protective gene, *HMOX1* has been reported to have protective effects in acute and chronic lung injury such as hyperoxia, hypoxia, ischemia, and hypertension [16,17,18,19]. Although there have been many reports on the structure and function of *HMOX1* gene, its transcriptional regulation mechanism has not been fully elucidated. Promoters play a major role in the regulation of gene transcription, and in-depth study of gene promoters is of great significance in explaining biological growth and development and disease defense [20].

Therefore, in the present study, with the Hezuo Tibetan pigs as the research object, we aim to explore the transcriptional regulation mechanism of the *HMOX1* gene, and understand the promoter structure and transcriptional activity of Tibetan *HMOX1* gene. First, the mRNA expression level of Hezuo Tibetan pig *HMOX1* gene in different tissues was quantified. Then, we cloned 5’ flanking promoter region and analyzed the sequence of *HMOX1* gene. In addition, a series of deletion recombinant plasmids were constructed for double-luciferase activity analysis. The transcription factor binding sites of the core promoter region was predicted by bioinformatics software. To further investigate the regulatory mechanism of this gene, site-directed mutation and RNA interference experiments were used to verify the critical transcription factors that regulate the *HMOX1* gene. This study will provide a reference for further studying the function of *HMOX1* gene and lay a foundation for research transcriptional regulation mechanism of pig *HMOX1* gene. 

## 2. Materials and Methods 

### 2.1. Ethics Statement

All animal experimental procedures used in this study were carried out following the guidelines of the China Council on Animal Care, and the protocols were subject to approval by the ethics committee of College of Animal Science and Technology, Gansu Agricultural University (approval number 2006-398).

### 2.2. Quantitative Real-Time PCR Analysis of Gene Expression Patterns

Fourteen tissues (heart, liver, spleen, lung, kidney, stomach, duodenum, jejunum, ileum, cecum, colon, rectum, psoas, dorsal longest muscle) were collected respectively from three adult Hezuo Tibetan pig in Hezuo, Tibetan Autonomous Prefecture of Gannan, China. Total RNA was extracted with *TransZol* Up reagent (TransGen Biotech, Beijing, China). For reverse transcriptional reaction, the first-strand cDNA was obtained according to the instructions of PrimeScript™ RT reagent kit with gDNA Eraser (TaKaRa, Dalian, China). The qRT-PCR was analyzed using TB Green Premix Ex Taq II (Tli RNaseH Plus) quantitative kit (TaKaRa, Dalian, China) and each reaction mixture was incubated in the Roche LightCycler 480 II instrument (Roche Applied Science, Penzberg, Germany). The primer information in this experiment is shown in Table 1. Relative expression levels of all genes were normalized with of β-actin (ACTB) expression, and the 2^−ΔΔCt^ method was used to calculate gene expressions [21].

### 2.3. HMOX1 Gene Promoter Region Cloning and Bioinformatics Analysis

The genomic DNA was extracted from Hezuo Tibetan pig blood using the TIANamp Blood DNA Kit (TIANGEN, Beijing, China), which as a template for PCR amplifications. An ~2 kb promoter region of the Hezuo Tibetan pig *HMOX1* gene (NCBI accession no. NM_001004027.1, region from 1952829 to 1954953) was amplified using specific primers (HMOX1-P-F/R, Table 1). Transcription initiation site (TSS) was predicted using Neural Network Promoter Prediction (http://www.fruitfly.org/seq_tools/promoter.html) online software, AliBaba2.1 [22] (http://gene-regulation.com/pub/programs/alibaba2/index.html) and JASPAR [23] (http://jaspar.genereg.net/) were used to predict the potential transcription factor binding sites. The CpG islands were predicted using MethPrimer [24] (http://www.urogene.org/methprimer/). NCBI Conserved Domain Database (CDD) (https://www.ncbi.nlm.nih.gov/cdd) was used to analyze the conserved domain. Nucleotide homology analysis was performed using the core promoter sequence of *HMOX1* gene, mainly including six species: pig (*Sus scrofa*, NW_018084968.1), human (*Homo sapiens*, NC_000022.11), mouse (*Mus musculus*, NC_000074.6), cattle (*Bos taurus*, NC_037332.1), sheep (*Ovis aries*, NC_040254.1), and horse (*Equus caballus*, NC_009171.3). MEGA7.0 software was used for multi-sequence alignment and homologous tree construction.

### 2.4. Cell Culture, Transfection, and Dual-Luciferase Reporter Assay

The 293T cells and Porcine alveolar macrophages (3D4/21) were purchased from BeNa Culture Collection (BNCC, Beijing, China). The cells were cultured in growth medium containing 90% DMEM (HyClone, New York, NY, USA), 10% fetal bovine serum (Invitrogen, Carlsbad, CA, USA), and 1% antibiotics (100 IU/ml penicillin and 100 μg/ml streptomycin) at 37 °C and 5% CO_2_. When the cells reached 70–80% confluence, cells were incubated in 24-well plates. For transfection, cells were transfected using Lipofectamine™ 2000 reagent (Invitrogen, Carlsbad, CA, USA). In order to determine the core promoter of *HMOX1* gene, a series of promoter fragments (−1878/+115, −1617/+115, −919/+115, −387/+115, −158/+115) were amplified through 5’ unidirectional deletion specific primers containing *Nhe* I and *Xho* I restriction enzyme sites, respectively. The PCR products were cloned into pGL3-basic luciferase reporter vector (Progema, Madison, WI, USA) using T4 DNA Ligase (TaKaRa, Dalian, China). After enzyme digestion and sequencing identification, the recombinant plasmids were extracted using EndoFree Mini Plasmid Kit II (TIANGEN, Beijing, China), and named pGL3-1878/+115, pGL3-1617/+115, pGL3-919/+115, pGL3-387/+115, and pGL3-158/+115, respectively. In order to verify the promoter activity of different fragments, each recombinant plasmid (800 ng) was co-transfection with internal vector pRL-TK (20 ng) using Lipofectamine™ 2000 reagent according to the manufacturer’s protocol. After 48 h post-transfection, the luciferase activity was detected using the Dual Luciferase Reporter Assay System (Promega, Madison, WI, USA), the pGL3-basic vector was considered as a negative control. Each group was performed three times.

### 2.5. Site-Directed Mutagenesis

The AliBaba2.1 and JASPAR online software were used to predict the transcription factor binding sites in core promoter region. In the present study, we mutated the potential transcription factor binding sites for WT1, Sp1, C/EBPα, and c-Ets-1 with the corresponding primers (Table 1) using Fast Site-Directed Mutagenesis Kit (TIANGEN, Beijing, China) according to the instruction manual.

### 2.6. RNA Interference

The siRNAs used in this experiment were designed and synthesized by GenePharma Company (Shanghai, China). The si-NC was regarded as a negative control. The interference sequences were shown in Table 1. Porcine alveolar macrophages (3D4/21) were cultured in 24-well plates and siRNAs (50 nM) co-transfected with pGL3-387/−158 plasmid (500 ng) according to the method mentioned before. After 48 h post-transfection, the relative mRNA expression and luciferase activity were measured as described above.

### 2.7. Statistical Analysis

The IBM SPSS Statistics (version 21.0) were used to analyze the relative mRNA expression levels of *HMOX1* gene in different tissues of Hezuo Tibetan pig, and the Duncan method was used to compare multiple groups. Independent sample *t*-test was used to analyze the relative luciferase activity among different promoter fragments. GraphPad Prism 8.0 software (Huntington, West Virginia, USA) was used for drawing. All values in this study were expressed as the mean ± standard deviation (SD). * indicates *p* < 0.05, ** indicates *p* < 0.01, n = 3.

## 3. Results

### 3.1. Tissue Expression Analysis of mRNA

Total RNA was extracted from fourteen tissues (heart, liver, spleen, lung, kidney, stomach, duodenum, jejunum, ileum, cecum, colon, rectum, psoas muscle, longissimus dorsi), qRT-PCR was used to analyzed the mRNA expression profiles of different tissues. Compared with the expression of *HMOX1* in the duodenum, the relative expression level of adults Hezuo Tibetan pig was shown in Figure 1. The *HMOX1* gene expression in the spleen was highest (*p* < 0.01) compared to other tissues. In addition, the liver, lung, kidney, and heart tissues also had the higher expression levels.

### 3.2. Promoter Region Cloning and Bioinformatics Analysis

Based on the pig *HMOX1* gene sequence published by GenBank database (NM_001004027.1), a 2125 bp 5’-flanking sequence spanning nucleotides from −1911 to +214bp was amplified (Figure 2A). We used NCBI-BLAST program to perform a sequence alignment analysis between the cloned Hezuo Tibetan pig promoter region sequence and 2125 bp sequence upstream of the pig *HMOX1* gene in NCBI, and found that the sequence similarity was 96.63%. In addition, we summarized the gene structure of *HMOX1* and found that it contains five exons in genome, and the length is about 9 kb. The mRNA (NM_001004027.1) transcript length is 1552 bp. In addition, an open reading frame (ORF) of 867 bp, which encoded 288 amino acids (aa) with one conserved domain, the Heme_oxygenase domain (in aa 11 to 216) was identified (Figure 2B). The results are shown in. The transcription start site (TSS) was predicted through Neural Network Promoter Prediction and the adenine residue (A) proximal to 5’ untranslated regions was verified and designated as +1 (Figure 2D). The CpG island was predicted by MethPrimer online software and found that two CpG islands located in 197 bp (−1516~−1320) and 225 bp (−164~+61) at promoter region (Figure 2C).

### 3.3. Promoter Activity Analysis and Core Promoter Region Identification

Through enzyme digestion and sequencing, we successfully constructed five recombinant vectors as shown in Figure 3B. Then we measured luciferase activity to find the core promoter region of *HMOX1* gene. The results indicated that pGL3-387/+115 had the highest transcription activity compared to other regions and significantly increase compared to the pGL3-158/+115 recombinant plasmid (*p* < 0.01) (Figure 3A). In order to identify the minimum core promoter region, we constructed the pGL3-387/−158 recombinant vector by subcloning, and the luciferase activity was shown that the pGL3-387/−158 significantly increase compared to pGL-158/+115 (*p* < 0.01). This suggested that −387/−158 as the core promoter region of *HMOX1* gene (Figure 3C). We further analyzed transcriptional regulatory elements in this region using AliBaba2.1 and JASPAR. As a result, four essential transcription factors—WT1, Sp1, C/EBPα, and c-Ets-1—were predicted in the core promoter region (Figure 4A). In addition, multiple alignments of core promoter region sequences between six species (pig, human, mouse, cattle, sheep, and horse). The results showed that the WT1, Sp1, C/EBPα, and c-Ets-1 regulatory elements were conserved in livestock (Figure 4B).

### 3.4. Roles of WT1, Sp1, C/EBPα, and c-Ets-1 in Transcriptional Regulation of HMOX1 Gene

In order to explore the role of WT1, Sp1, C/EBPα, and c-Ets-1 in the regulation transcription of *HMOX1* gene, four recombinant plasmids with 4 bp point mutations in the WT1, Sp1, C/EBPα, and c-Ets-1 binding sites in pGL-387/-158 were constructed. Then, the four mutant vectors were transfected into 293T and 3D4/21 cells to detect the luciferase activity, respectively. The results showed that the luciferase activity of the mutated WT1, Sp1, and C/EBPα were significantly decreased (*p* < 0.01), while the luciferase activity of mutated c-Ets-1 had no significant changes (Figure 5A). To further validate the roles of WT1, Sp1, and C/EBPα binding sites in the core promoter region of the *HMOX1* gene, the selected transcription factors were silenced through siRNAs. First, the interference efficiency of the siRNAs (si-WT1, si-Sp1, and si-C/EBPα) were detected through negative control siRNA after transfection 24 h. The results showed that these siRNAs significantly reduced (*p* < 0.01) the mRNA expression levels of the WT1, Sp1, and C/EBPα as compared to the negative control (Figure 5B). In addition, the relative mRNA expression level of *HMOX1* gene had significantly decrease (*p* < 0.05 and *p* < 0.01) by WT1, Sp1, and C/EBPα inhibition, respectively (Figure 5C). Further co-transfection of si-WT1, si-Sp1, or si-C/EBPα with pGL-387/−158 also reduced luciferase activity (Figure 5D). In this context, we see that WT1, Sp1, and C/EBPα can promote the transcriptional activity of promoter and increase the expression of *HMOX1* gene. These three transcription factors may be bind to the core promoter region of *HMOX1* gene and play a role in activating *HMOX1* gene transcription. 

## 4. Discussion

Heme oxygenase (HO) family has three important members, including inducible *HMOX1* and constitutive *HMOX2* and *HMOX3* [25,26]. The *HMOX1* gene, also known as heat-shock protein 32 (HSP32), plays an important role in protecting the body from a variety of stimuli and pathological conditions [27,28,29]. Studies have reported that vascular endothelial growth factor (VEGF) can regulate the expression of *HMOX1* gene and play an important regulatory role in angiogenesis [30,31]. Wang et al. [32] found that *HMOX1* has a protective effect in fish hypoxic cells. In addition, Gou et al. [33] reported that increasing hemoglobin content is a key molecular mechanism of hypoxia adaptation in Tibetan chicken embryos. These studies suggest that the HO family genes may play an important protective role in the body’s anti-inflammatory, antioxidant stress, and hypoxia injury activities.

In this study, the expression profiles of *HMOX1* gene in Hezuo Tibetan pigs were analyzed in 14 tissues and found that the expression level of *HMOX1* gene in spleen tissue was the highest, followed by liver, lung, and kidney. This results are consistent with the previous findings of Kovtunovych et al. [34] on mice and Park et al. [35] on transgenic pigs. At present, bioinformatics analysis and construction of luciferase reporter gene vectors are common methods to study promoter activity and core regulatory regions [36]. In this work, we successfully constructed recombinant vectors with different deletion lengths in the *HMOX1* gene promoter region and detected luciferase activity after transfection of 293T cells and 3D4/21 cells. It was found that the *HMOX1* gene promoter region -387/-158 had the highest activity, indicating that there were important positive regulatory transcription elements in this region. Subsequently, bioinformatics predicted that the promoter region −387 / −158 of *HMOX1* gene contains crucial transcription factor binding sites such as WT1, Sp1, C/EBPα, and c-Ets-1. Further combined with site-directed mutagenesis and RNA interference experiments, we have preliminarily confirmed that WT1, Sp1, and C/EBPα can activate the promoter activity of *HMOX1* gene. Transcription factors can bind to specific DNA sequences in the gene promoter region to control the transcription and expression of genes, thereby regulating various physiological activities of the body [36]. The siRNAs can specifically inhibit or degrade the expression of homologous mRNA, thereby mediating gene post-transcriptional silencing [37]. In this study, si-WT1, si-Sp1, and si-C/EBPα were designed and synthesized. After transfected into 3D4/21 cells, it was found that si-WT1, si-Sp1, and si-C/EBPα could effectively inhibit the expression of *HMOX1* mRNA level, and can reduce the *HMOX1* gene promoter activity. The previous study found that WT1 transcription factor (WT1) can play a role in heart and blood vessel formation [38]. It was also regarded as a key element in acute myeloid leukemia [39], and WT1 overexpression was an independent positive prognostic factor in adult B-cell acute lymphoblastic leukemia patients [40]. The transcription factor Sp1 (Sp1) can regulate the expression of multiple genes [41], which can play a role in the body’s antioxidant stress [42]. Under hypoxic and ischemic conditions, regulating the expression of Sp1 can regulate the activation mechanism of coagulation in rats with hemorrhagic shock [43]. Sp1 can promote TGF-β1 expression and activate the SMAD2 pathway, the SP1/TGF-β1/SMAD2 pathway may enhance angiogenic processes in preosteoblasts [44]. Deshane et al. [45] showed that Sp1 also regulated the expression of *HMOX1* gene in human kidney cells. CCAAT/enhancer binding protein α (C/EBPα) can play a role in cell proliferation, apoptosis, inflammation, and other responses. According to previous study, the C/EBPα can play a role in the development of lipogenesis [46] and acute myeloid leukemia [47]. Liu et al. [48] found that C/EBPα was a key target gene of DNA methyltransferase 1 (Dnmt1) downstream, which can play a crucial role in maintain hematopoietic stem and progenitor cells in zebrafish. It has also been reported that C/EBPα can increase the expression of *HMOX1* gene under the induction of peptidoglycan [49]. Taken together, the transcription factors WT1, Sp1, and C/EBPα can regulate the expression of multiple genes and participate in a variety of biological processes related to hematopoietic function. Therefore, we speculate that transcription factors WT1, Sp1, and C/EBPα may be significant factors in regulating *HMOX1* gene, this research may provide a positive reference for further studying the transcriptional regulation mechanism of *HMOX1* gene.

## 5. Conclusions

In conclusion, in the present study, we analyzed *HMOX1* gene expression profiles and gene structure. We also cloned 5’ promoter region and predicted the transcription initiation site of Hezuo Tibetan pig *HMOX1* gene. In addition, we discovered that the core promoter region and three transcription factors WT1, Sp1, and C/EBPα are likely to play an important role in the expression of *HMOX1* gene in Hezuo Tibetan pig. Our results will provide a basic information for further research the transcriptional regulation mechanism of *HMOX1* gene. 

## Figures and Tables

**Figure 1 genes-11-00352-f001:**
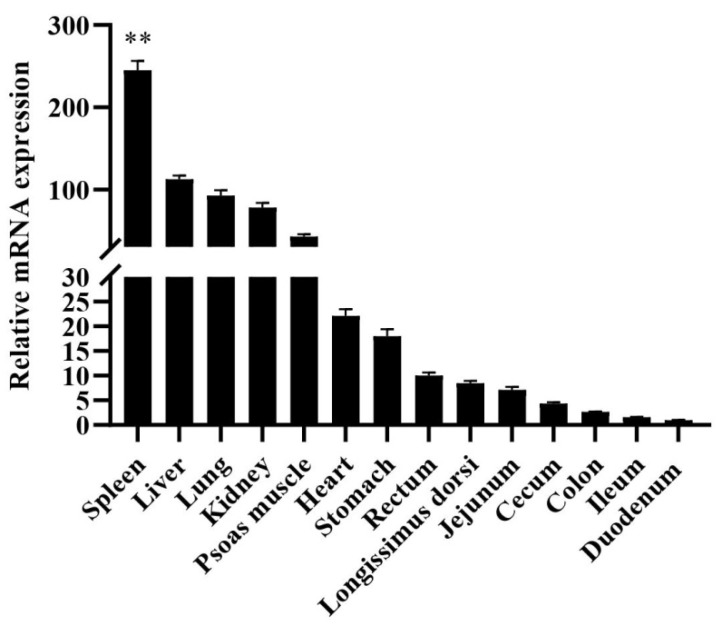
Relative expression patterns of Hezuo Tibetan pig *HMOX1* gene in different tissues. The result was normalized with *β-actin* gene and relative to gene expression in the duodenum group. ** indicates *p* < 0.01.

**Figure 2 genes-11-00352-f002:**
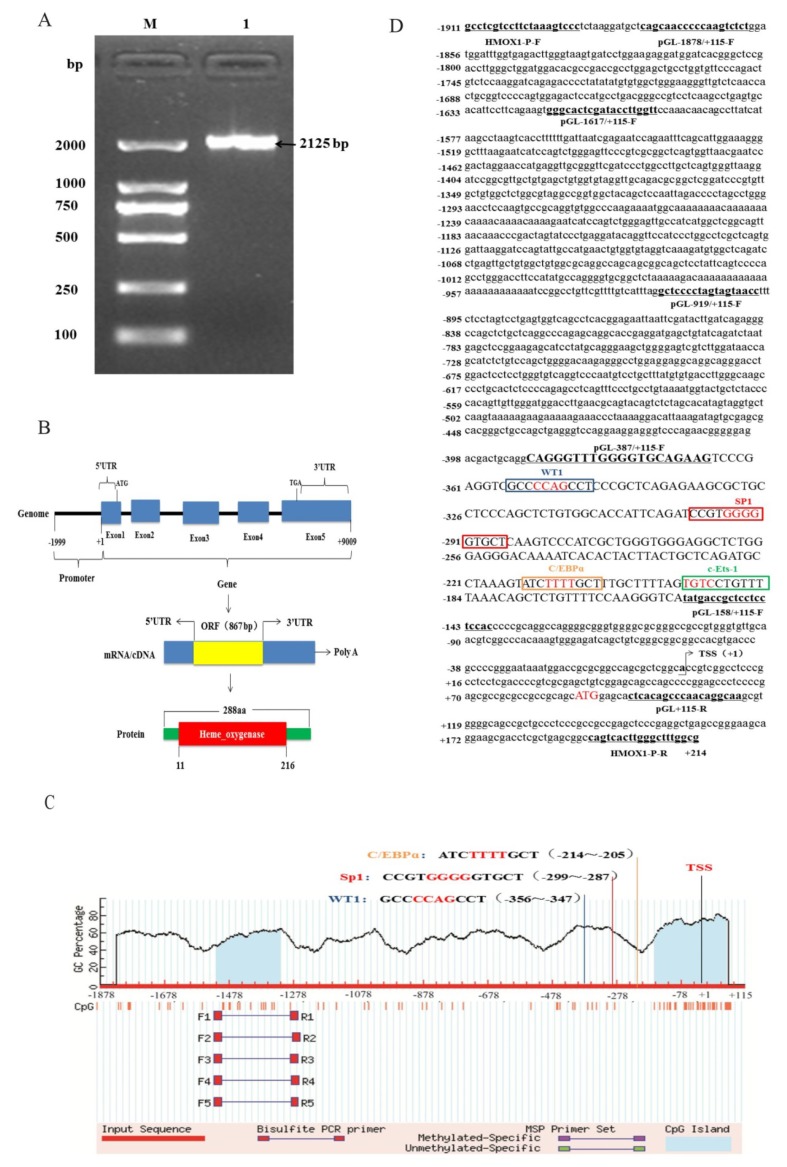
Promoter region cloning and bioinformatics analysis. (**A**) PCR amplification product of *HMOX1* gene promoter region. M, DL2000 DNA marker; 1, PCR product. (**B**) Genomic structure of *HMOX1* gene. (**C**) The predict results of CpG island in *HMOX1* gene. (**D**) The promoter sequence information of *HMOX1* gene. The box represents the transcription factor binding sites, and the 4 bp core binding sites are shown in red. The underline represents the amplification primer and the arrow represents the transcription start site (TSS).

**Figure 3 genes-11-00352-f003:**
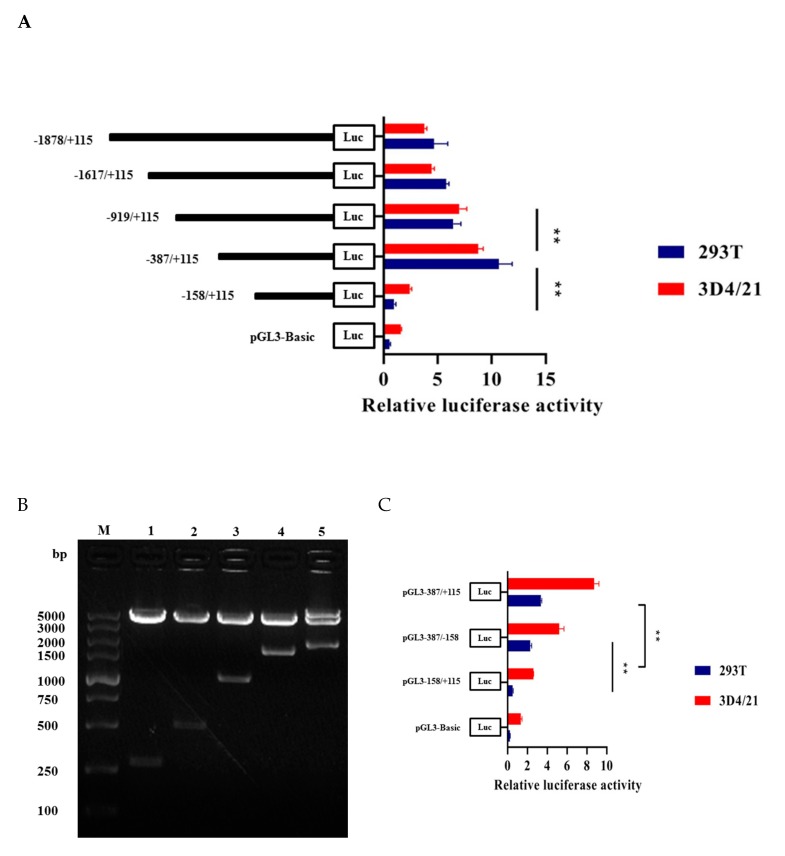
Construction of different deletion vectors of *HMOX1* gene promoter region and luciferase activity analysis. (**A**) Luciferase activity analysis of promoter with different deletion vectors (pGL3-1878/+115, pGL3-1617/+115, pGL3-919/+115, pGL3-387/+115, pGL3-158/+115), pGL3-Basic as a negative control. (**B**) Double enzyme digestion identification of different deletion vectors. M, DL5000 DNA marker, lanes 1–5 are different deletion fragments. (**C**) Analysis of luciferase activity in the core promoter region. All ** represents *p* < 0.01.

**Figure 4 genes-11-00352-f004:**
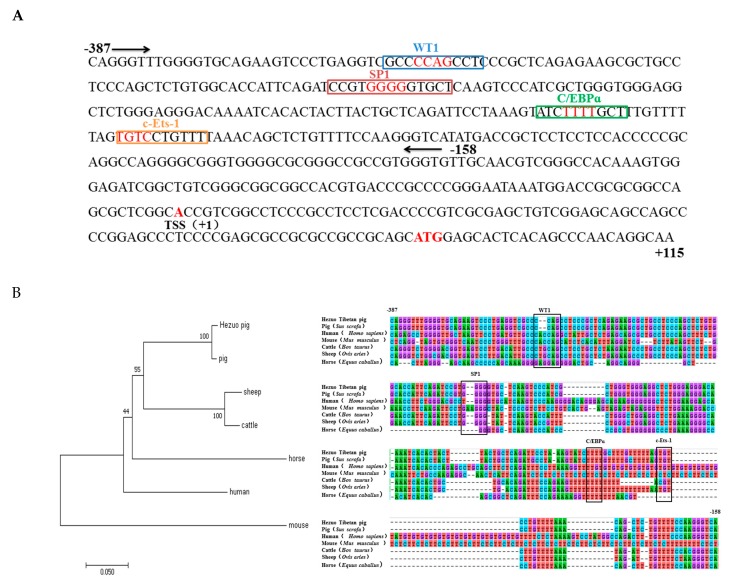
Transcription factor prediction and multi-species homology analysis in the core promoter region. (**A**) Transcription factor prediction of core promoter region. Four essential transcription factors WT1, Sp1, C/EBPα, and c-Ets-1 were predicted in −387/−158, the location of the 4 bp core binding is indicated in red. (**B**) Sequence alignment and phylogenetic construction of core promoter regions in six species. The box represents the 4 bp core binding site.

**Figure 5 genes-11-00352-f005:**
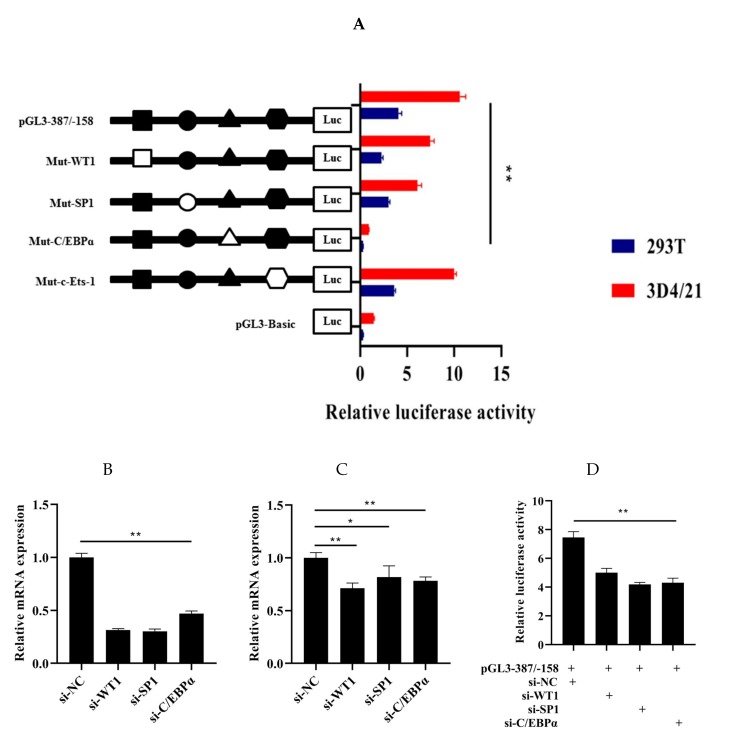
Role of WT1, Sp1, and C/EBPα in the transcription regulation of *HMOX1* gene. (**A**) Luciferase activity analysis with site-directed mutagenesis of WT1, Sp1, C/EBPα, and c-Ets-1 sites. Solid and hollow represent wild-type and mutant-type, respectively. (**B**) Interference efficiency of WT1, Sp1, and C/EBPα. (**C**) The mRNA expression level of *HMOX1* gene by inhibition with si-WT1, si-Sp1, and si-C/EBPα. (**D**) Luciferase activity after co- transfection of si-WT1, si-Sp1, and si-C/EBPα with pGL3-387/−158. (* indicates *p* < 0.05, ** indicates *p* < 0.01).

**Table 1 genes-11-00352-t001:** Primer information in this study

Name	Primer Sequence(5’–3’)	Tm (°C)	Length	Region
HMOX1-P	F: GCCTCGTCCTTCTAAAGTCCC	56	2125 bp	−1911/−1892
	R: CGCCAAAGCCCAAGTGACTG			+214/+195
HMOX1-RT	F: GACATGGCCTTCTGGTATGGG	60	141 bp	348–368
	R: CATGTAGCGGGTGTAGGCGT			488–469
WT1-RT	F: GGTGTCTTCAGGGGCATTCA	60	102 bp	1127–1146
	R: ACACATGAAGGGGCGTTTCT			1228–1209
Sp1-RT	F: TGTCTCTGGTGGGCAGTATG	60	133 bp	601–620
	R: TTGCCCATCAACCGTCTGG			733–715
C/EBPα-RT	F: GGCAAAGCCAAGAAGTCGGT	60	124 bp	976–995
	R: TCTGTTGAGTCTCCACGTTGC			1099–1079
β-actin-RT	F: ATATTGCTGCGCTCGTGGT	60	148 bp	142–160
	R: TAGGAGTCCTTCTGGCCCAT			289–270
HMOX1-P1	F: CTA**GCTAGC**TATGACCGCTCCTCCTCCAC	60	273 bp	−158/+115
HMOX1-P2	F: CTA**GCTAGC**CAGGGTTTGGGGTGCAGAAG	60	502 bp	−387/+115
HMOX1-P3	F: CTA**GCTAGC**GCTCCCCTAGTAGTAACCTGC	64	1034 bp	−919/+115
HMOX1-P4	F: CTA**GCTAGC**GGGCACTCGATACCTTGGTT	60	1732 bp	−1617/+115
HMOX1-P5	F: CTA**GCTAGC**CAGCAACCCCCAAGTCTCT	62	1993 bp	−1878/+115
HMOX1-R	R: CCG**CTCGAG**TTGCCTGTTGGGCTGTGAG			
WT1-m	F: GAAGTCCCTGAGGTCGCC**GGTC**CCTCCCGCTCAGAGAAGC		229 bp	−387/−158
	R: GCTTCTCTGAGCCAGCGG**CCAG**GGAGGGCTCAGGGACTTC			
Sp1-m	F: GGCACCATTCAGATCCGT**AAAA**GTGCTCAAGTCCCATCGC		229 bp	−387/−158
	R: GCGATGGGACTTAGGCA**TTTT**CACGAGCTGAATGGTGCC			
C/EBPα-m	F: TCAGATTCCTAAAGTATC**AAAA**GCTTTGTTTTTAGTGTCC		229 bp	−387/−158
	R: GGACACTAAAAATCATAG**TTTT**CGAAACTTAGGAATCTGA			
c-Ets-1-m	F: CTTTTGCTTTGTTTTTAG**ACAG**CTGTTTTAAACAGCTCTG		229 bp	−387/−158
	R: CAGAGCTGTTTAAAAATC**TGTC**GACAAAACAAAGVAAAAG			
si-WT1	F: GGGCUGCAAUAAGAGAUAUTT			
	R: AUAUCUCUUAUUGCAGCCCTT			
si-Sp1	F: GCGGAUCUGCAGUCCAUUATT			
	R: UAAUGGACUGCAGAUCCGCTT			
si-C/EBPα	F: ACGAGACGUCCAUCGACAUTT			
	R: AUGUCGAUGGACGUCUCGUTT			
si-NC	F: UUCUCCGAACGUGUCACGUTT			
	R: UUAACUCAUCGCUUCUUGCTT			

**GCTAGC: **The the bold and underlined represent restriction site. **GGTC: **the bold and underlined represent site-directed mutation site.

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
