# Peer review of "Transcriptional Regulation of HMOX1 Gene in Hezuo Tibetan Pigs: Roles of WT1, Sp1, and C/EBPα"

_genes, 2020, doi:10.3390/genes11040352_

Round 1
Reviewer 1 Report
1- HMOX1 is a well-known gene in pig (NCBI and Ensembl annotations). In this manuscript, authors tried to predict the structure of this gene in Hezuo Tibetan Pigs. However, there is no sentence describing the difference between this breed and the breed used in current pig genome assembly/annotation throughout the manuscript in the introduction section.
2- Lines 149-159, Please describe the possible similarity/differences between the predicted HMOX1 gene structure in this experiment with structures reported for this gene in Ensembl or NCBI annotations.
3- Figure 1, please change PSOAS to PSOAS muscle
Reviewer 2 Report
Review #: genes-749697
Transcriptional Regulation of HMOX1 Gene in Hezuo Tibetan Pigs: Roles of WT1, Sp1 and C/EBPα
General comments:
The manuscript by Wei Wang and collaborators analyze HMOX1 gene expression profiles and gene structure. They have also identified the core promoter region of this gene and three transcription factors (WT1, Sp1 and CEBPA) involved in the transcriptional regulation of this gene. In this study, the Hezuo Tibetan pigs as the research object, in order to explore the transcriptional regulation mechanism of HMOX1 gene, and understand the promoter structure and transcriptional activity. However, it is not sufficiently clear, that the function and regulation of this gene allow explaining or clarifying its implication in the adaptation of the Tibetan pigs to the high altitude and hypoxia environment. I have only minor suggestions. Before publication, some minor changes are recommended.
ABSTRACT:
The last sentence of the abstract should be modified because in principle this sentence is too categorical as it does not give a fully provide reference to the role of this gene in the adaptation of the Tibetan pig to high altitude and hypoxia environment. The last sentence is not adequately supported by the data in the manuscript.
INTRODUCTION:
HMOX1 attenuates adipogenecity and obesity but in the introduction, I have not seen any reference to this, which would also have great importance in the adaptive capacity of Tibetan pigs (for example Park et al., 2015).
In my opinion, I do not see clearly the implication of the function and regulation of this gene in the adaptation of Tibetan pigs to extreme environmental conditions. Check the introduction to clarify this better.
MATERIAL AND METHODS:
L.79-80: You have only used one endogenous gene to normalize the expression data, this is not usual; normally two reference genes are used to normalize. Please explain in more detail how you have done it.
RESULTS:
Figure 1: “Longissimus dorsi” should be in italics
Figure 2: The font size is very small it is difficult to see for example in sections B,C y D
L.214-216: Please explain this phrase in more detail
DISCUSSION:
L.246-248: The discussion is not sufficiently justified. Some sentences are not properly justified.
L.259-266: This sentence is not sufficiently justified. Please provide more citations for this statement. The sentences should be re-written.
CONCLUSIONS:
L.273-274: Remove or change this sentence. The purpose of this study not provide a basis for exploring the high altitude and hypoxia adaptability of Tibetan pigs
